# Structural basis for perception of diverse chemical substances by T1r taste receptors

Nipawan Nuemket[1,2,*,†], Norihisa Yasui[1,*], Yuko Kusakabe[3], Yukiyo Nomura[1], Nanako Atsumi[1], Shuji Akiyama[4,5], Eriko Nango[2], Yukinari Kato[6,7], Mika K. Kaneko[7], Junichi Takagi[8], Maiko Hosotani[1] & Atsuko Yamashita[1,2]

The taste receptor type 1 (T1r) family perceives 'palatable' tastes. These receptors function as T1r2-T1r3 and T1r1-T1r3 heterodimers to recognize a wide array of sweet and umami (savory) tastes in sugars and amino acids. Nonetheless, it is unclear how diverse tastes are recognized by so few receptors. Here we present crystal structures of the extracellular ligand-binding domains (LBDs), the taste recognition regions of the fish T1r2-T1r3 heterodimer, bound to different amino acids. The ligand-binding pocket in T1r2LBD is rich in aromatic residues, spacious and accommodates hydrated percepts. Biophysical studies show that this binding site is characterized by a broad yet discriminating chemical recognition, contributing for the particular trait of taste perception. In contrast, the analogous pocket in T1r3LBD is occupied by a rather loosely bound amino acid, suggesting that the T1r3 has an auxiliary role. Overall, we provide a structural basis for understanding the chemical perception of taste receptors.

[1] Laboratory of Structural Biology, Graduate School of Medicine, Dentistry and Pharmaceutical Sciences, Okayama University, 1-1-1, Tsushima-naka, Okayama 700-8530, Japan. [2] RIKEN SPring-8 Center, 1-1-1, Kouto, Hyogo 679-5148, Japan. [3] Food Research Institute, NARO, 2-1-12, Kannondai, Tsukuba 305-8642, Japan. [4] Research Center of Integrative Molecular System (CIMoS), Institute for Molecular Science, National Institute of Natural Sciences, 38 Nishigo-Naka, Myodaiji, Okazaki 444-8585, Japan. [5] Department of Functional Molecular Science, The Graduate University for Advanced Studies (SOKENDAI), 38 Nishigo-Naka, Okazaki 444-8585, Japan. [6] New Industry Creation Hatchery Center, Tohoku University, 2-1 Seiryo-machi, Aoba-ku, Sendai 980-8575, Japan. [7] Department of Antibody Drug Development, Tohoku University Graduate School of Medicine, 2-1 Seiryo-machi, Aoba-ku, Sendai 980-8575, Japan. [8] Institute for Protein Research, Osaka University, 3-2 Yamadaoka, Osaka 565-0871, Japan. * These authors contributed equally to this work. † Present address: Protein Crystal Analysis Division, Japan Synchrotron Radiation Research Institute, 1-1-1, Kouto, Sayo 679-5198, Japan. Correspondence and requests for materials should be addressed to A.Y. (email: a_yama@okayama-u.ac.jp).

Taste sensation enables animals to detect certain chemical substances within foods, and evaluate whether they are nutritious or poisonous. The process is evoked by specific interactions between stimulants and taste receptors residing in the plasma membrane of taste cells in the taste buds of the oral cavity[1,2]. The taste receptor type 1 (T1r) family discerns 'palatable' tastes in nutrients, such as sugars and L-amino acids[3–5]. The family is conserved across in vertebrates, including fishes, birds, and mammals[6], and receptors function as constitutive heterodimers of T1r1–T1r3 and T1r2–T1r3 (refs 3,4). Ligand specificity is likely tuned to the diet of the animals. In humans and rodents, the T1r2–T1r3 heterodimer recognizes sweet substances like sugars, whereas the T1r1–T1r3 heterodimer samples umami (savory tastes) of L-amino acids including glutamate[3–5]. In contrast, in birds, a group generally lacking the *T1r2* gene, the T1r1–T1r3 heterodimer from insect-feeding species responds to L-amino acids, while that from a nectar-feeding species detects sugars[7].

The physiology of taste perception is embodied in the characteristics of T1r function. Many T1r receptors have broad ligand specificity: the human T1r2–T1r3 receptor reacts to mono- to oligosaccharides, artificial sweeteners without saccharide groups, some D-amino acids and even proteins[5], while the mouse T1r1–T1r3 receptor responds to various L-amino acids[4]. This contrasts with endogenous signalling, which generally recognizes specific chemical substances such as hormones, cytokines, and neurotransmitters. The taste perception through T1r even contrasts with another chemosensation, olfaction sensation, where >1 trillion stimuli are discriminated by combinations of ∼400 receptors[8]. Another feature of T1r receptors is their low affinity for taste substances present in high concentrations in the oral cavity (EC50 values of human T1r1–T1r3 and T1r2–T1r3 for glutamate and sucrose are ∼2.7 mM (ref. 9) and ∼41 mM (ref. 10), respectively).

Chemical recognition by T1r proteins is mainly achieved by their extracellular ligand-binding domains (LBDs). The T1r family belongs to the class C G-protein-coupled receptor (GPCR) family[11], which commonly possess a LBD, consisting of ∼500 amino acid residues, upstream of the heptahelical transmembrane region, on the extracellular side (Fig. 1a). Mutation and modelling/docking studies point to the LBDs of T1r2 and T1r1 of T1r heterodimers as perceiving most of major sweet and umami taste substances[9,12–17], except for the artificial sweetener cyclamate that targets the transmembrane (TM) domain of T1r3 (refs 12,18) and the sweet protein brazzein where there is an additional contribution by a cysteine-rich domain (CRD) of T1r3 downstream of the LBD[19]. A mechanism of signal transduction by T1r has been proposed based mainly on other class C GPCR LBDs[20,21]: ligand binding induces closure of the cleft between subdomains, LB1 and LB2, possibly accompanying dimer rearrangement[22], to effect a conformational change in the downstream transmembrane region and receptor activation and signal transmission to the heterotrimeric G-protein in the cytosol of taste cells. However, details, especially how diverse chemicals are recognized, still need to be elucidated.

In this study, we have addressed the structural basis of taste perception by T1r by crystallographic analysis of its LBDs. Structural analysis of T1r has so far been hampered by the difficulties in heterologous expression[23]. Although there are a couple of studies reporting *E. coli* expression of a LBD from a single T1r subunit using fusion[24] or refolding[25] strategies, there have been no reports of the successful preparation of T1rLBDs in the functional unit of heterodimer. Recently, we showed that the LBDs of T1r2-subtype a (T1r2a) and T1r3 heterodimer from medaka fish (*Oryzias latipes*) can be successfully expressed in insect cells, despite failures with those from other sources

that we tested, including human genes[22]. Fish possess multiple T1r2 genes, which show an almost equal degree of sequence identity to both T1r1s and T1r2s (31–34%), and which trichotomously branched off from mammalian T1r2s at the node close to T1r1s and T1r3s (ref. 26). Indeed, T1r2a of medaka fish, one of three T1r2 subtypes in the species, with heterodimeric partner T1r3, responds to several kinds of L-amino acids[27]. Therefore, medaka fish T1r2a–T1r3 may be a good representative member of T1rs and allow exploration of general characteristics, such as polyspecific recognition. Here we report the crystal structures of the T1r2a–T1r3LBD heterodimer, with a variety of bound amino acids, providing a basis for the understanding of the broad chemical recognition of taste receptors.

## Results

**Overall structure**. Although attempts to crystallize the T1r2a–T1r3LBD heterodimer alone were unsuccessful, co-crystallization with a Fab fragment (Fab16A), prepared from an antibody recognizing T1r2aLBD, yielded crystals, enabling structure determination at 2.2–2.6 Å resolution (Table 1 and Supplementary Figs 1, 2a,b, 3). The structure showed that T1r2aLBD and T1r3LBD have the architecture of the Venus-flytrap domain (VFTD; Fig. 1b), also found in other class C GPCR LBD structures such as metabotropic glutamate receptors (mGluRs)[20], GABAB receptor (GABABR)[21], and calcium-sensing receptor (CaSR)[28] (Supplementary Fig. 2c–e). The T1r2aLBD–T1r3LBD heterodimer takes a compact dimer arrangement, similar to the 'A'-state observed in the glutamate-bound mGluR1LBD structure[20], as expected from our previous study mainly based on small-angle X-ray scattering analysis[22] (Supplementary Fig. 2c,f, and Supplementary Table 1).

**Heterodimerization**. The structure provides a basis for the heterodimerization of T1r, which is required for normal taste receptor function[3–5,29]. First, distinctive intermolecular interactions at the loop regions are observed, where loop 1 (Ala46–Asp57) and loop 2 (Val116–Ala133) in T1r2a and loop 2 (Thr121–Asp140) in T1r3 alternately cross and fold over the LB1 domain in the other subunit, forming intermolecular main-chain hydrogen bonds (Fig. 1c and Supplementary Fig. 4a). Notably, Cys132 in T1r3 of loop 2 forms an intermolecular disulfide bridge with Cys344 in T1r2a of loop 3 (Gly336–Ser354). This disulfide bridge was confirmed by mutational analysis: the mutant C132A in T1r3 or C344A in T1r2a displayed markedly decreased dimer formation (Supplementary Fig. 4b). In the cases of mGluRs and CaSR, cysteine residues in loop 2, such as Cys140 in mGluR1 and a pair of Cys129 and Cys131 in CaSR, form intermolecular disulfide bond(s) between subunits of the homodimer[30,31]. On the other hand, no intermolecular disulfide bridge was actually expected in T1r heterodimers, since T1r1 and T1r2 have no cysteine residues in loop 2 (Supplementary Fig. 1). However, the characteristic configuration of the loop regions is such that intermolecular disulfide bridge formation with loop2 in T1r3 and loop 3 in T1r2a is allowed. The interaction between cysteines of loop 2 and 3 in the different subunits means that the disulfide-mediated dimer stabilization is unlikely between T1r1- or T1r2-homodimers, even if such oligomers do exist. Second, the intermolecular interface between the LB1s of T1r2aLBD and T1r3LBD is complementary: both surfaces are hydrophobic, and there are several coupling points that could facilitate heterodimeric interactions, such as between Asp103 in T1r2a and Lys158 in T1r3 (Fig. 1d). These non-covalent heterodimeric interactions together with the covalent disulfide bridge described

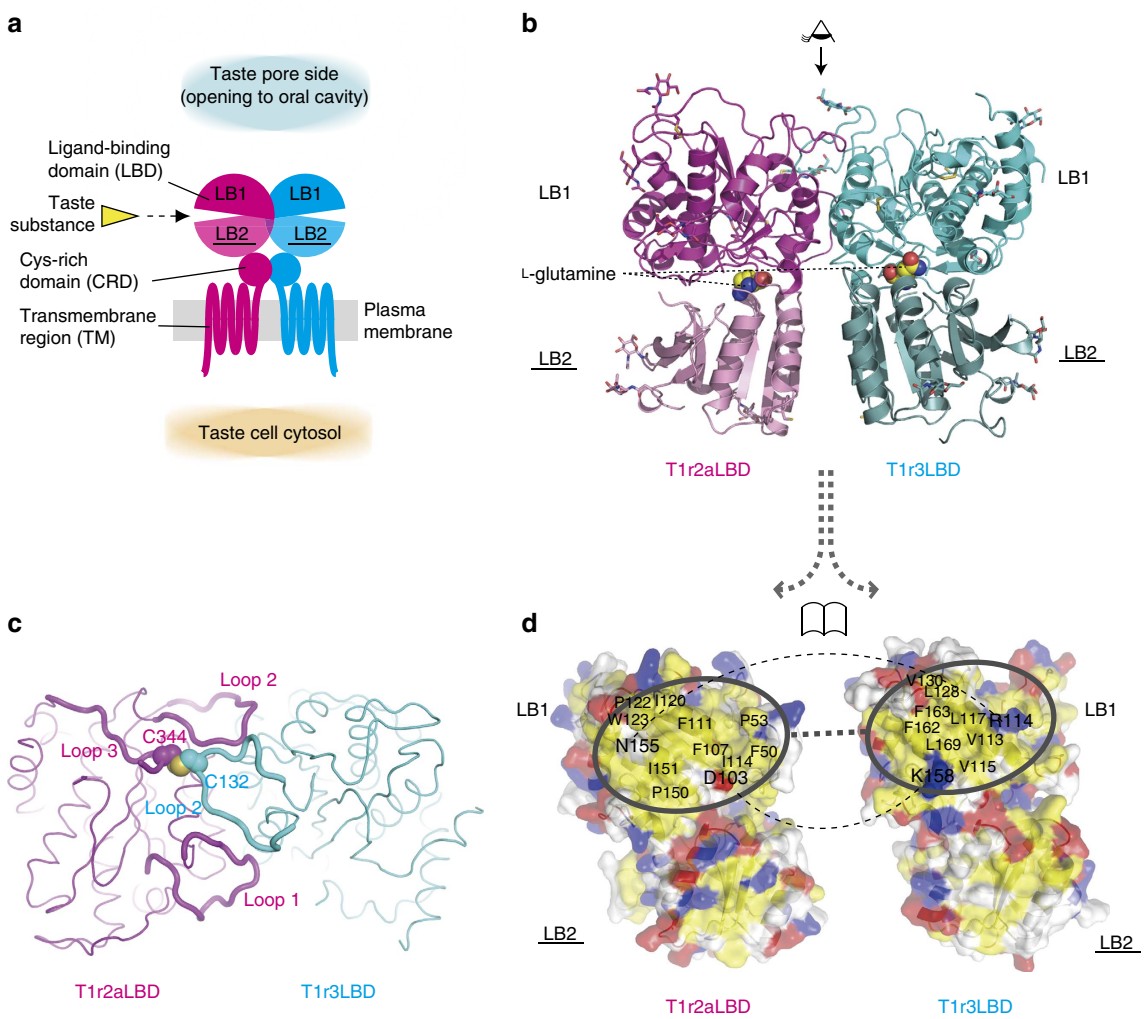

**Figure 1 | Crystal structure of the medaka fish T1r2a–3LBD heterodimer.** (**a**) Schematic drawing of T1r taste receptor heterodimer. (**b**) The crystal structure of the T1r2a–T1r3LBD heterodimer. The bound ligand molecules (L-glutamine, shown as spheres) and post-translational modifications (disulfide bonds and glycosylation, shown in sticks, Supplementary Fig. 1) are also shown. (**c**) Intermolecular interaction mediated by the loop regions at LB1, viewed from the point shown in panel b, depicted as an eye above the structure. (**d**) Intermolecular interfaces T1r2aLBD and T1r3LBD. Hydrophobic, acidic, basic, and polar uncharged amino acid residues are coloured in yellow, red, blue and white, respectively.

above must shift the equilibrium of T1rLBDs towards heterodimers, rather than homodimers or dissociated states.

**Ligand recognition by T1r2a subunit.** The T1r2a–T1r3 heterodimer from medaka fish is reported to be a receptor for amino acids[27], which are considered to be preferred taste substances for medaka fish[32]. We found that a variety of L-amino acids, such as glutamine, alanine, arginine, glycine, and glutamate, but not D-glutamine, induced FRET signal elevation of fluorescence-labelled LBDs, which we had previously found to correlate with LBD ligand binding[22] (Fig. 2a and Supplementary Table 2). Some of them, such as glutamine, alanine, and arginine, also induced responses in the full-length receptor according to $Ca^{2+}$-imaging (Fig. 2b and Supplementary Table 2). The two amino acids displaying weak affinities to LBD, glycine and glutamate, gave poor or no significant receptor responses up to 10 mM, which is presumed to be below their saturation taking into account the several fold higher concentration ranges for the responses than those for the FRET changes observed on the

former three amino acids (Supplementary Table 2), probably due to the assay method differences[22]. Higher concentrations up to saturation were not investigated to avoid non-specific reactions. Nevertheless, the results indicate that T1r2a–3LBD recognizes a broad spectrum of L-amino acids where the α-substituent group differs in size, charge, and hydrophobicity. The observed concentration ranges for the receptor responses fit the typical taste thresholds for L-amino acids exhibited by fishes (0.1–100 mM)[33], which are much higher than those of specific receptors for endogenous signalling molecules, such as mGluR ($EC_{50}$ 4.6~22 μM for L-glutamate response)[34].

In the crystal structures, these various L-amino acids are bound to T1r2aLBD in the cleft between LB1 and LB2 (Fig. 1b and Supplementary Fig. 2b). The T1r2aLBD structures adopt a closed conformation as judged by structure superposition to LBDs of mGluRs and $GABA_BR$[20,21] (Supplementary Fig. 2c), as well as domain motion analysis, which indicates that LB2 is rotated ~28° towards LB1 compared to the open conformation of the mGluR1 structure (PDB ID 1EWK, B chain). The degree of this rotation compares well with the domain motion between closed

**Table 1 | Data collection and refinement statistics.**

|  | L-Gln bound | L-Ala bound | L-Arg bound | L-Glu bound | Gly bound | SeMet bound |
|---|---|---|---|---|---|---|
| *Data collection* |  |  |  |  |  |  |
| Space group | $P2_1$ | $P2_1$ | $P2_1$ | $P2_1$ | $P2_1$ | $P2_1$ |
| Cell dimensions |  |  |  |  |  |  |
| $a$ (Å) | 99.7 | 99.4 | 98.9 | 99.6 | 98.5 | 99.5 |
| $b$ (Å) | 116.4 | 117.5 | 113.7 | 116.1 | 112.7 | 115.8 |
| $c$ (Å) | 129.5 | 130.0 | 128.6 | 129.6 | 128.7 | 129.6 |
| $\beta$ (°) | 92.2 | 91.9 | 92.2 | 91.7 | 92.2 | 91.9 |
| Resolution (Å) | 50.0–2.20 | 50.0–2.20 | 50.0–2.60 | 50.0–2.60 | 50.0–2.60 | 50.0–3.10 |
| $R_{sym}$ (%)* | 7.8 (70.5) | 8.2 (58.0) | 12.5 (38.0) | 10.4 (50.0) | 9.1 (39.1) | 14.3 (47.7) |
| $I/\sigma(I)$ * | 15.6 (2.1) | 18.7 (1.8) | 11.9 (2.1) | 12.2 (1.8) | 10.7 (1.8) | 12.3 (3.0) |
| Completeness (%)* | 97.1 (96.2) | 95.3 (97.1) | 95.9 (94.1) | 96.1 (95.9) | 98.0 (97.2) | 98.8 (99.3) |
| Redundancy* | 3.3 (3.0) | 3.5 (3.5) | 2.8 (2.3) | 3.1 (2.8) | 3.4 (3.0) | 5.5 (5.3) |
|  |  |  |  |  |  |  |
| *Refinement* |  |  |  |  |  |  |
| Resolution (Å) | 50–2.2 | 50–2.2 | 50–2.6 | 50–2.6 | 50–2.6 |  |
| No. reflections | 142,729 | 144,159 | 84,554 | 86,084 | 84,790 |  |
| $R/R_{free}$ (%) | 17.3/22.7 | 19.5/24.4 | 18.5/27.0 | 15.3/22.8 | 15.4/23.0 |  |
| No. atoms |  |  |  |  |  |  |
| Protein | 20,320 | 20,577 | 20,366 | 20,218 | 20,467 |  |
| Sugar | 378 | 322 | 364 | 389 | 361 |  |
| Ligand | 40 | 24 | 64 | 40 | 20 |  |
| Ion | 6 | 4 | 8 | 5 | 9 |  |
| Water | 775 | 745 | 363 | 417 | 407 |  |
| B-factors |  |  |  |  |  |  |
| Protein | 52.10 | 48.64 | 44.70 | 43.82 | 37.64 |  |
| Sugar | 87.90 | 71.66 | 76.40 | 71.57 | 72.73 |  |
| Ligand | 43.99 | 36.19 | 53.60 | 53.19 | 32.72 |  |
| Ion | 60.43 | 38.56 | 53.93 | 72.99 | 39.74 |  |
| Water | 47.57 | 44.47 | 37.13 | 38.47 | 36.51 |  |
| R.m.s. deviations |  |  |  |  |  |  |
| Bond lengths (Å) | 0.009 | 0.008 | 0.011 | 0.009 | 0.012 |  |
| Bond angles (°) | 1.158 | 1.100 | 1.224 | 1.208 | 1.236 |  |

R.m.s., root mean square.
*Values in parentheses refer to data in the highest resolution shells.

and open conformations of other class C GPCR LBDs (21–36°, Supplementary Data 1). Not only T1r2aLBD but also the overall heterodimer structures were very similar irrespective of the bound ligand identity (Cα rmsds of 0.4 ∼ 0.7 Å).

The α-amino and carboxyl groups and the Cα and Cβ atoms of the amino acid ligands are located at almost the same positions in the binding site in T1r2a, and form hydrogen bonds with the protein (Fig. 2c–h, and Supplementary Fig. 5), including to the main-chain amide and side-chain hydroxyl groups of Ser142 and Ser165, and the main chain carbonyl group of Gly163, all residues in LB1 (Fig. 2c). Ser142 and Ser165 are conserved in amino-acid sensing taste receptors T1r1 and most class C GPCRs (Supplementary Fig. 1), and they indeed form a similar pattern of hydrogen bonds with the amino and carboxyl groups of L-glutamate and GABA in mGluRs and GABA$_B$R, respectively (Supplementary Fig. 5). Mutation of Ser165 to isoleucine, an amino acid observed in human T1r2 (Supplementary Fig. 1), or alanine resulted in the loss of or a weakened response to L-amino acids, down to the level of the non-specific reaction with D-amino acids (Fig. 2i and Supplementary Fig. 6), although the cell surface expression of the full-length receptors, as well as secretion and heterodimerization of the LBDs, remain (Supplementary Fig. 7). The results are consistent with previous studies showing that the same mutant of human T1r1, S172A, does not respond to L-glutamate and other amino acids[9,14]. The results indicate that the α-amino and carboxyl groups of the amino acid ligand are key for recognition by T1r2aLBD, primarily by the LB1 region, and for eliciting a receptor response.

In contrast, the α-substituent groups of the bound amino acids, such as the 2-carbamoylethyl group in L-glutamine, have no direct hydrogen bonds with the receptor. However, numerous water molecules surround these substituent groups, and the ligands are thus in hydrated states. Indirect, water-mediated hydrogen bonds are apparent, such as with the carbonyl group of Ala263 or the carboxyl group of Asp288 and Asp289, in the LB2 region. These interactions likely help closing the cleft between LB1 and LB2. Notably, higher affinity ligands, such as L-glutamine and alanine, are surrounded by more ordered water molecules, implying that the binding site recognizes the ligand together with the structured water in the hydration shell (Fig. 2c–h). This is in striking contrast to receptors with a high ligand specificity, such as mGluRs and GABA$_B$R, in which direct hydrogen bonds and salt bridges are formed with the residues in the LB2 region of the receptor and the ligand molecules (Supplementary Fig. 5). The way in which the hydration shell participates in ligand recognition as observed in T1r2aLBD appears suitable for polyspecific ligand perception, since replacement of water molecules, rearrangement of the hydrogen-bond network, and charge screening by the hydration shell allow diverse ligands to be accommodated in the binding pocket without varying the conformation of the protein.

Broad recognition of a hydrated ligand is achieved by several distinct structural characteristics in the ligand-binding pocket in T1r2a. The space that accommodates the ligand is much larger than any of the ligands (Fig. 2j), and there is no evidence of significant induced fit on binding (Fig. 2c–h), which allows

accommodation of ligand *plus* hydration shell. The ligand-binding space is not completely occluded from the solvent, existing as a 'back room' with ~7 Å height and ~16 Å depth, restricted by Ile64, Pro66, and Lys265 at ~8 Å distance from the carbamoyl group of the bound glutamine. In contrast, the space

for ligand binding in mGluR is smaller, with ~7 Å height and ~11.5 Å depth, restricted by Tyr74 and Arg323 at ~2.5 Å distance from the γ-carboxyl group of the glutamate (Fig. 2k). The T1r2a pocket is constituted by a number of aromatic residues, such as Phe140, Phe213, Phe262 and Phe365 (Fig. 2c–h).

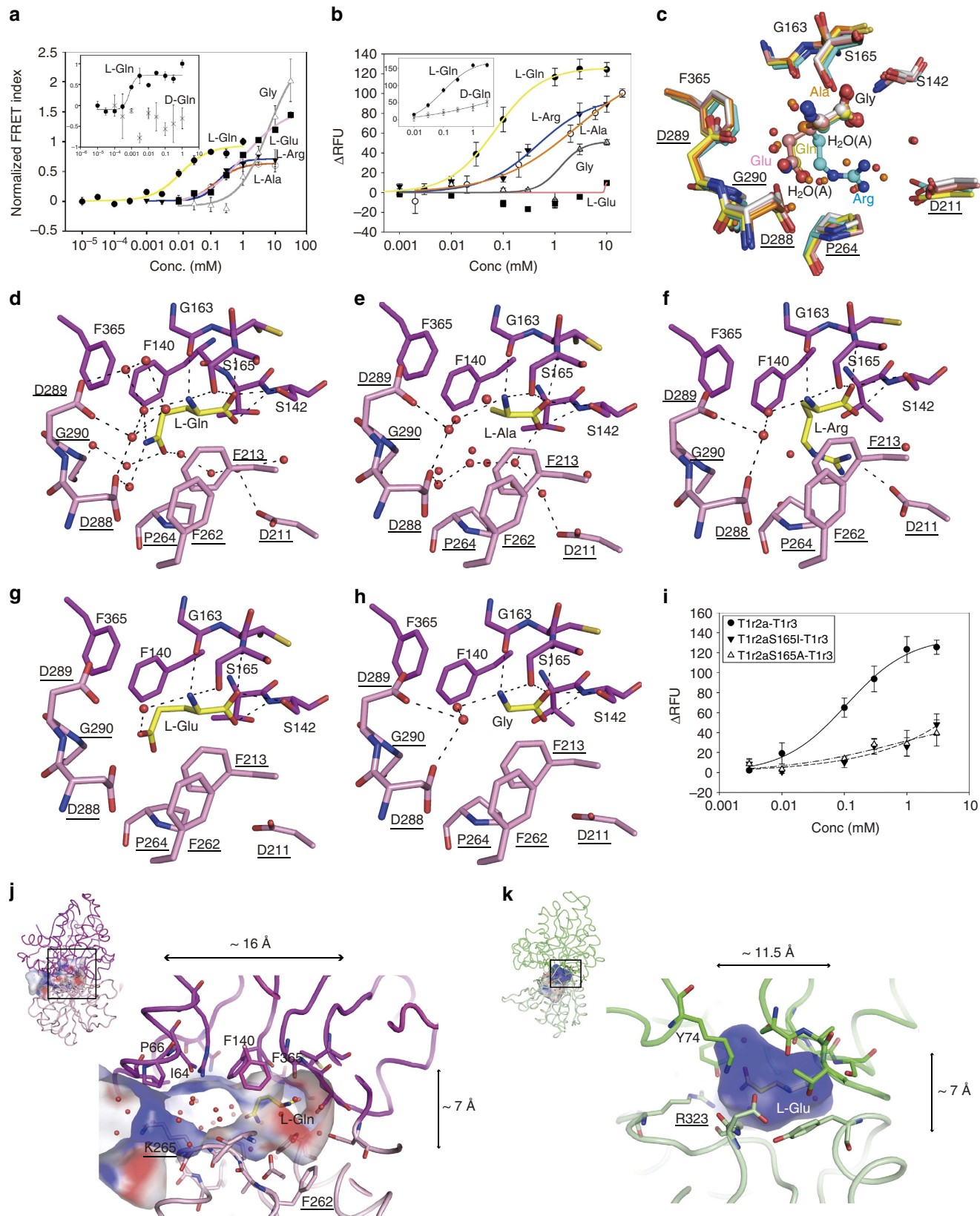

These residues may serve as a platform for water-aromatic interactions through OH-π, lone pair-π and CH-O interactions[35,36], as well as helping to keep the large, yet restricted, space for ligand binding due to their structural rigidity. Numerous aromatic amino acid residues are often observed in the drug-binding sites of multidrug efflux transporters with broad ligand specificity[37–39]. Furthermore, the surface of the T1r2a pocket is a mosaic of negatively, positively and uncharged regions, in contrast to the uniformly positively charged surface of the mGluR1 pocket (Fig. 2j,k). All of these features facilitate broad rather than specific substrate recognition, and there is a trade-off between low affinity and high affinity.

**Ligand binding at T1r3 subunit.** The T1r2a–3LBD crystals display an unanticipated additional electron density in the cleft of T1r3, at the site corresponding to the amino acid binding pocket observed in T1r2a (Fig. 3a). This confounded the previous assumption that the LBDs of T1r2 and T1r1 are solely responsible for ligand binding[12,14], and we came to the conclusion that the LBD of T1r3 binds the same amino acid as that of T1r2a, based on the following observations. The shape of the omit map in the T1r3 cleft in each crystal resembles the amino acid added for crystallization (Fig. 3a and Supplementary Fig. 8a–d). We further confirmed amino-acid binding at T1r3 by crystallographic analysis in the presence of L-selenomethionine, which results in a noticeable anomalous peak derived from the selenium atom (5.5 σ) at the T1r3 binding site, as well as at T1r2a (6.0 σ, Supplementary Fig. 8e,f). Previous studies also reported that the recombinant mouse and human T1r3LBDs are capable of binding sweet substances[24,25].

Similar to T1r2aLBD, T1r3LBD adopts a closed conformation (Supplementary Fig. 2c) as judged by structure superposition of other class C GPCR LBDs and domain motion analysis, which showed that the LB2 is ~25° rotated towards LB1 compared to the open conformation of the mGluR1 structure (PDB ID 1EWK, B chain, Supplementary Data 1). It contrasts with GABA$_B$R, another heterodimeric class C GPCR, in which subunit GBR2 does not bind ligands and adopts an open conformation[21]. Nevertheless, T1r3 alone exhibits no obvious response to amino acids in the tested concentration ranges (Supplementary Fig. 8g), again suggesting the dominance of T1r2a for receptor response.

In the T1r3-binding site, the α-amino and carboxyl groups of the ligand form hydrogen bonds with Ser150 and Thr173 in LB1, similar to those observed in T1r2a and other class C GPCRs structures (Fig. 3a and Supplementary Fig. 5). In addition, the α-amino group forms a water-mediated hydrogen bond with the hydroxyl group of Ser300 in LB2. Since there are no other significant interactions with LB2, this single hydrogen bond might induce closure of the cleft between LB1 and LB2, irrespective of the α-substituent group of the bound amino acid. The ligand binding pocket in T1r3 is structurally distinct from

that in T1r2a, as well as that in mGluR1; the cavity is ~7.6 Å high but with no restriction in depth between the solvent and the α-substituent group (Fig. 3b). There is no obvious interaction of the α-substituent with LB2, even through water molecules, thus the binding site cannot serve as a determinant of receptor specificity. When we mutated Ser300 to glutamate, an amino acid observed in human T1r3 (Supplementary Fig. 1), receptor responses appeared wild-type like (Fig. 3c and Supplementary Table 2). Ligand binding to T1r3 seems mute in terms of receptor response, although an interaction of the glutamate at the Ser300 with the α-amino group cannot be ruled out.

The relevance of the structural characteristics of the ligand binding pockets of the LBDs of T1r2a and T1r3 for ligand binding was evaluated by calculating ΔG values from their crystal structures. The differences of the estimated ΔG values (ΔΔG) between the various amino-acid binding and L-glutamine binding to T1r2a are in the ~2 kcal mol$^{-1}$ range (Fig. 3d), and remarkably, correlate with those derived from experimental binding assays for T1r2a–3LBD by FRET (Fig. 2a and Supplementary Table 2), except for L-arginine, which exhibits a unique binding configuration of the guanidinopropyl group, forming a salt bridge with Asp211 (Fig. 2c,f). In contrast, binding to T1r3 showed uniform ΔΔG values in a low range (< ~0.3 kcal mol$^{-1}$, Fig. 3d) and there is no apparent correlation with the experimental binding results. These results corroborate the above conclusions that ligand binding to T1r2a is responsible for ligand specificity of the T1r receptor and binding to T1r3 is non-specific.

## Discussion

This study reveals the structural basis of the broad amino acid recognition by LBDs of the T1r taste receptor. The crystal structures displayed that a wide range of amino acids, irrespective of their physicochemical properties, are accommodated to the ligand binding site in hydrated states. The T1r2aLBD takes the primary role in T1r2a–T1r3 heterodimer receptor stimulation, both in terms of affinities for agonist binding and resultant receptor responses, as judged by a structural comparison with T1r3LBD as well as mutation analyses. These observations are in line with the notion that the unique component of the T1r receptors, T1r1 or T1r2, is responsible for recognition and responses to taste substances, despite the fact that the common component, T1r3, has a similar ligand binding site. Therefore, the similar characteristics may be conserved in T1rs in other species and explain why sweet and umami taste modalities are versatile yet discriminative. T1r3 likely plays a subsidiary role in the receptor function of T1r, such as inter-subunit conformational coupling and G-protein coupling[24], or membrane trafficking of T1r heterodimers[40]. In addition, the ligand binding properties of T1r3 observed in this study may substantiate previous observations that T1r3 alone responds to

**Figure 2 | Amino-acid recognition by T1r2aLBD. (a)** Dose-dependent FRET signal changes of the T1r2aLBD-Cerulean and T1r3LBD-Venus heterodimer for amino acid binding. Data points represent mean and s.e.m. of three technical replicates. **(b)** Dose-response curves for various amino acids by the full-length T1r2a–T1r3 receptor in HEK293 cells, monitored as an elevation of intracellular Ca$^{2+}$ concentration. Data points represent mean and s.e.m. of six, eight, six, six and four technical replicates for the Gln, Arg, Ala, Gly and Glu responses, respectively. For **a,b**, the insets are the curves for L- and D-glutamine. (three and four technical replicates for **a,b**, respectively.) **(c–h)** Close-up view of the T1r2a ligand-binding site of the L-glutamine-, L-alanine-, L-arginine-, L-glutamate- and glycine-bound structures. **(c)** super-imposition of all structures, with the water molecules observed at the binding site of the glutamine-bound (red) and alanine-bound (orange) structures plotted, and those specifically observed on the alanine-bound structure labelled as 'H$_2$O(A)'. **(d)** L-Glutamine-bound structure, **(e)** L-alanine-bound structure, **(f)** L-arginine-bound structure, **(g)** L-glutamate-bound structure, **(h)** glycine-bound structure. **(i)** L-Glutamine responses of the wild-type T1r2a–T1r3, and T1r2a:S165I–T1r3 and T1r2a:S165A–T1r3 mutants, monitored as an elevation of intracellular Ca$^{2+}$ concentration. Data points represent mean and s.e.m. of 4 technical replicates. **(j,k)** The ligand-binding pockets observed on T1r2a **(j)** and mGluR1 (PDB ID 1EWK, A chain) **(k)** crystal structures. The electrostatic potentials at ± 20 kTe$^{-1}$ were mapped on the surfaces. In all panels, the residues at LB2 are underlined.

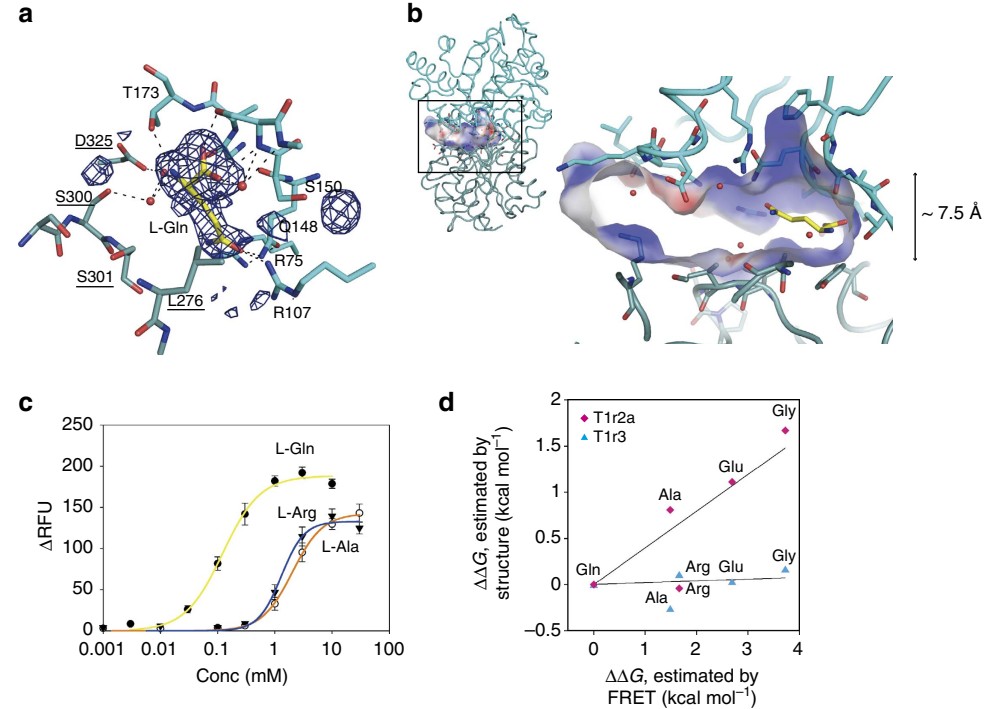

**Figure 3 | Non-specific amino acid recognition by T1r3LBD. (a)** Close-up view of the T1r3 ligand-binding site of the L-glutamine-bound structure. Simulated annealing-omit electron density map (3.0 σ) is also shown. The residues at LB2 are underlined. (**b**) The ligand-binding pockets observed on T1r3, with the electrostatic potential at ± 20 kT e$^{-1}$ mapped on the surfaces. (**c**) Responses of the T1r2a–T1r3:S300E mutant to various amino acids, monitored as an elevation of intracellular Ca$^{2+}$ concentration. Data points represent mean and s.e.m. of 11, 12 and 6 technical replicates for the Gln, Arg and Ala responses, respectively. (**d**) The $\Delta G$ difference ($\Delta\Delta G$) between L-glutamine and other amino-acid binding of T1rLBD. The $\Delta\Delta G$ values estimated by FRET (Fig. 2a and Supplementary Table 2) were plotted on those estimated by the structures of T1r2a (magenta diamonds) and T1r3 (cyan triangles) binding sites.

chemicals in certain situations[29,41,42], which may exist in tongue and palate epithelium, as well as in various tissues throughout the body, expressing T1r3 and not T1r1 or T1r2 (refs 3,43).

The crystal structures of T1r2a–3LBD indicate that the various amino acids induce almost the same closed LBD conformation (Supplementary Fig. 2b and Supplementary Data 1). This structural observation is consistent with the fact that various amino acids induce the receptor responses (ref. 27 and Fig. 2b), which are considered to correlate with the conformational change of LBD. On the other hand, this is apparently not in accordance with the observation that they elicit different maximum FRET changes at saturating concentrations (larger changes with glutamate and glycine compared with glutamine, Fig. 2a). Recently, single-molecule FRET studies of mGluRLBDs suggested that there are not only two simple 'active' and 'relaxed' states but multiple conformational states, and the efficacies of agonists are determined by their properties to shift the conformational equilibria of the LBD population, yielding differences in fractional occupancy of each state[44,45]. Our observed difference in maximum FRET changes in T1rLBD among the amino acids may reflect different occupancies of multiple conformational states, while crystallization apparently traps the single conformational state in this study. A similar situation was observed in the crystallographic structures of a class C GPCR, mGluR II LBD, where a variety of agonists with different efficacies, such as full and partial agonists, induce almost the same LBD conformation[46]. In addition, effects of the ligands on the other regions of the receptor, such as CRD and TM, are obscure. Future studies on the conformational dynamics of T1r in the presence of various ligands and how the movements correlate with receptor response, as well as structural analysis of the full-length receptor, may be revealing in this regard.

## Methods

**Measurement of the response of T1r2 and T1r3 to ligands.** The conformational change of T1r2a–T1r3LBD on ligand binding was analysed by Förster resonance energy transfer (FRET) measurement. The heterodimer of the T1r2aLBD (M1-S474)—Celurean fusion protein (T1r2aL-Ce) and the T1r3LBD (M1-S491)—Venus fusion protein (T1r3L-Ve) was stably expressed by S2 cells (Invitrogen) and purified by ANTI-FLAG M2 affinity gel (SIGMA) by use of the FLAG-tag at their C termini, as previously described[22]. The sample was excited at 433 nm, and the emission at 526 and 475 nm was recorded with a FluoroMax4 spectrofluorometer (Horiba) at room temperature, except for the comparative analysis of L- and D-glutamine, shown in Fig. 2a inset, performed at 283 K. The FRET index (Intensity at 526 nm/Intensity at 475 nm) change in the presence of a ligand was normalized with the value in the presence of 1 mM L-glutamine, and was plotted against ligand concentration. The titration curves were fitted to the Hill equation by using KaleidaGraph (Synergy Software). For Fig. 3d, $\Delta G$ for each amino acid binding was estimated assuming that $K_d$ is equal to the EC$_{50}$ value for FRET change, and the temperature is 298 K, and the differences between that for L-glutamine binding ($\Delta\Delta G$ in the horizontal axis) were plotted against those calculated from the crystal structures as described below in the 'Crystallography' section.

The full-length receptor responses of T1r2a–T1r3 to ligands were analysed as previously described[22]. Briefly, the Flip-In 293 cell line (Life Technologies) was used for stable expression of T1r2a, T1r3, and Gα16-gust44 (ref. 47). The Ca$^{2+}$ flux assays were performed using a FLEX station 3 (Molecular Devices, LLC) for the cells loaded with 100 µl of Hank's balanced salt solution (Sigma-Aldrich), containing 5 µM of the calcium indicator dye Fluo8 NW (AAT Bioquest). The stimulation was performed by adding 25 µl of 5 × concentrated solutions of taste substances, using a pipette. The intensity of the response was represented as $\Delta$RFU (delta relative fluorescence unit) defined as maximum fluorescence value for the taste substances minus that for the Hank's Balanced Salt Solution, and was plotted versus the ligand concentration. The concentration-response curves were fitted to the Hill equation, using SigmaPlot software (Systat Software).

Mutations were introduced using QuikChange (Agilent Technologies) or PrimeStarMax (TAKARA BIO, only for T1r2a:S165I) with primers summarized in Supplementary Table 3.

On the basis of our experiences, more than two independent experiments and three technical replicates (three sample sets) for each experiment are considered to be adequate to grasp the properties of ligand binding to LBD and cell

responses[22,40]. Therefore, at least two independent measurements were performed in this study, except for the inset in Fig. 2a with a single analysis. The Gln- and Ala-binding and responses by the wild-type T1r2a-3(LBD) are independent repeats of those reported previously[22].

**Antibodies and Fab fragment preparation and analyses.** Mouse anti-T1r2a–3LBD monoclonal antibodies were produced by Mikuri Immunological Lab. Co. (Yao, Osaka, Japan), using the standard protocol. The purified T1r2a–3LBD sample was used as the antigen. Antibodies recognizing conformational epitopes were selected as follows. First, the hybridomas exhibiting high titers in a conventional ELISA were further screened by native versus unfolded ELISA, as reported previously[48] with some modifications. Briefly, the T1r2a–3LBD samples under native conditions and unfolded conditions, prepared by a 10 min incubation at 368 K in the presence of 6 M guanidine-HCl and 3 mM DTT followed by 6-fold dilution, were coated on the Ni-NTA HisSorb 96-well plate. TBS-DDM (20 mM Tris-HCl, 150 mM NaCl, pH 7.5, 0.01% dodecylmaltoside) with (for native) or without (for unfolded) 100 mM L-alanine was used as the wash buffer, and the results were detected with the ABTS solution (Pierce) using a Varioskan Flash plate reader (Thermo Scientific). Secondly, selected hybridomas were further screened by FSEC[22,23] for their ability to elicit a shift in elution peak position of T1r2a–3LBD-GFP on mixing with the culture media. Thirdly, the hybridomas were further characterized by western blotting, and those exhibiting low or no signals for the denatured T1r2a–3LBD protein were selected. The cloning of the selected hybridoma, the large-scale antibody production, and the purification were performed by Mikuri Immunological Lab. Co. by conventional methods.

The Fab fragment was generated by IgG digestion with Immobilized-Papain (Pierce), according to the manufacturer's protocol, except for an increase of L-cysteine concentration in digestion buffer to 100 mM. The digested IgG was further purified by chromatography on Protein A Sepharose (GE Healthcare), pre-equilibrated with Protein A IgG binding buffer (Pierce). The digested Fab fragment was further purified by ion chromatography on a Mono S 5/50 GL column (GE Healthcare). The bound Fab was eluted with a linear gradient of NaCl up to 300 mM over 20 column volumes in 20 mM sodium acetate, pH 5.0. The purified Fab was stored in 50 mM Tris-HCl, pH 7.6, 150 mM NaCl at 193 K until used.

The N-terminal amino acid sequences for the heavy and light chains of the anti-T1r2a–3LBD mAb (clone 16A, mouse IgG1:kappa) were determined to be EVQLQQSGPE and DIVLTQSPAS, respectively, by Edman sequencing. Total RNA was prepared from the hybridoma cells, using an RNeasy PLUS Mini Kit (Qiagen). The initial cDNA strand was synthesized using the SuperScript III First-Strand Synthesis System (Thermo Fisher Scientific) via a priming oligo-dT, according to the manufacturer's instructions. PCR amplification was performed with oligonucleotide mixtures of the degenerate primer of the N-terminal amino acid sequence and the constant region primer for the heavy ($\gamma$) and light ($\kappa$) chains[49,50]. PCR reactions were performed with HotStar *Taq* polymerase (Qiagen). The PCR products were purified, subcloned into the pCR4-TOPO vector (Thermo Fisher Scientific), and subjected to nucleotide sequencing. The primer sequences are summarized in Supplementary Table 3.

**Crystallography.** The protein sample for crystallization was expressed and purified as described previously[22], except for some minor modifications. Briefly, the C-terminal FLAG-tagged T1r2aLBD (M1-S491)–3LBD (M1-S491) heterodimer was stably expressed using S2 cells in ExpressFiveSFM (Life Technologies) supplemented with 5 µM kifunensine for five days. After the recovery of the protein from the culture medium with the ANTI-FLAG M2 affinity gel (SIGMA), the tag region was cleaved by factor Xa (Novagen) 0.09 U per 1 µg of protein at 283 K overnight. Factor Xa and the undigested protein were removed by Hitrap Benzamidine (GE Healthcare) and ANTI-FLAG resin, respectively. The glycosyl chains were trimmed to single *N*-acetylglucosamine residues by His-tagged Endo H (400 µg per 1 mg protein) at 283 K overnight. After the Endo H removal by Ni-NTA agarose (Qiagen), the protein was mixed with the purified Fab at a 1:1.5 molar ratio and incubated for 1 h at 277 K before the final purification by size exclusion chromatography (HiLoad Superdex 200, 16/60 (GE Healthcare)), using buffer A (20 mM Tris, 5 mM L-glutamine, 2 mM CaCl$_2$, 0.3 M NaCl, pH 8.0) as the running buffer. For the preparation of the protein samples bound to L-arginine, L-alanine, L-glutamate or glycine, the T1r-Fab mixture was applied to a size exclusion chromatography column, equilibrated with buffer A in which the glutamine was substituted with 0.1 M of each amino acid. For the selenomethionine (SeMet)-bound sample, the sample buffer was substituted with buffer A containing 50 mM L-SeMet by PD-10 (GE Healthcare). Before crystallization, the NaCl and Tris concentrations in the buffer were decreased to 50 and 10 mM, respectively using a Vivaspin 20 (Sartorius).

The T1r2a–3LBD-Fab complex crystals were obtained by the sitting-drop vapor diffusion method together with a microseeding technique, to induce crystal growth and improve the crystal quality. One µl of $\sim$ 12 mg ml$^{-1}$ protein sample was mixed with 1 µl of the reservoir solution containing 12$\sim$13% (w/v) PEG 1500, 3% (v/v) PEG400 and 0.1 M MES, pH 6.0, and 0.2 µl of seeding soup, and incubated at 293 K. The seeding soup was prepared by Teflon seed beads (Hampton Research), according to the manufacturer's recommendations.

The crystals were cryoprotected by gradually changing the mother liquor to the solution containing 0.1 M MES, pH 6.0, 50 mM NaCl, 17% PEG1500, 5% PEG400,

2 mM CaCl$_2$, 10% glycerol, and either 5 mM L-glutamine, 0.1 M L-alanine, 0.1 M L-arginine, 0.1 M L-glutamate, 0.1 M glycine, or 50 mM L-SeMet, and flash-cooled by plunging into liquid nitrogen. The X-ray diffraction data were collected at the wavelength of 1.0 Å, except for the Se-Met bound crystal collected at 0.979 Å for anomalous signal detection, at the SPring-8 beamline BL41XU, using a PILATUS3 6M detector (DECTRIS). The Photon Factory beamline BL-5A, with an ADSC Quantum 315r detector, was also used at the initial stage of the data collection. The data were processed with HKL2000 (ref. 51).

The structure of the L-glutamine-bound form of T1r2a–3LBD was solved by the molecular replacement method with the program PHASER[52], using the structures of a Fab fragment (PDB ID: 1A6T)[53] and a single protomer of the mGluR3LBD (PDB ID: 2E4U)[46] as the search models. The structure model was first constructed with BUCCANEER[54], followed by ARP/wARP[55]. The model was manually rebuilt with COOT[56], and refined with REFMAC[57], the PDB_REDO server[58], and PHENIX[59]. The bound ions, such as Na$^+$, Cl$^-$, and Ca$^{2+}$, were assigned based on the automatic placement by PHENIX[59] or visual inspection in terms of difference Fourier peaks and the structural characteristics of the binding sites. In the refined models, residues in the favoured, allowed, and outlier regions of the Ramachandran plots are: 96.2, 3.3, 0.6% for the Gln-bound form; 96.5, 3.2, 0.3% for the Ala-bound form; 95.2, 4.5, 0.3 for the Arg-bound form; 95.8, 3.8, 0.4% for the Glu-bound form; 96.6,3.1, 0.3% for the Gly-bound form. The structures of the two complexes in the asymmetric unit were almost identical, and we selected chains A, B, H and L, from one of the two complex models constructed in the region with better quality electron density, as representatives and describe them in the text.

The comparison between the crystal structure and the small-angle X-ray scattering curves analysed previously[22] was performed with CRYSOL[60]. The domain motion analysis was performed with DynDom[61]. In the case of analysis of two structures with sequence identity less than 40%, the amino acid sequences of the structures were mutated to that of mGluR1 using CHAINSAW[62] based on structure-based sequence alignment prepared by PROMALS3D[63], and the modified structures were subjected to the DynDom analysis. The properties of the ligand binding cavities were analysed by HOLLOW[64], PDB2PQR[65] and APBS[66]. For Fig. 3d, the $\Delta G$ for each amino acid binding was estimated from the crystal structure of L-glutamine bound T1r2a–3LBD by replacing the bound L-glutamine with other amino acids using the PositionScan option in FoldX[67], and the differences between that for L-glutamine binding were plotted as $\Delta\Delta G$ in the vertical axis. The structural figures were prepared with Pymol (Schrödinger) and LigPlot$^+$ (ref. 68).

**Immunoprecipitation and Western blotting.** Transient expression of T1rLBD proteins in S2 cells was carried out by the calcium phosphate method as described previously[22], or by using FuGENE HD (Roche) with 1 µg each T1r2aLBD and T1r3LBD expression vectors for $1 \times 10^6$ cells according to the manufacturer's protocol. The cells were cultivated at 300 K for 4 days. Mutations were introduced using the QuikChange method (Agilent Technologies).

The cell culture supernatant was mixed with ANTI-FLAG M2 affinity gel (SIGMA) and rotated at 277 K. After washing with 20 mM Tris, pH 7.4, containing 150 mM NaCl, the proteins retained on the beads were eluted with $2 \times$ SDS–PAGE sample buffer (100 mM Tris, 2% SDS, 10% (v/v) glycerol, 0.002% bromophenol blue, pH6.8), followed by heating at 368 K for 5 min. The eluents were divided into two equal parts, further incubated at 368 K for 5 min with or without 100 mM DTT, and subjected to SDS-PAGE followed by electrophoretic transfer onto membranes, using an iBlot apparatus (Life Technologies). T1rLBDs were detected using anti-DDDDK tag-HRP (1:2,000) (Cat. # PM020-7, Medical and Biological Laboratories) and Immobilon Western Chemiluminescent HRP Substrate (Millipore). The images were obtained using a ChemiDoc Imager (Bio-Rad). The uncropped original images of the blots are shown in Supplementary Fig. 9.

**Multicolour FSEC.** The wild-type and mutant T1r2aL-Ce and T1r3L-Ve heterodimer proteins were prepared as described above. The samples were loaded on a SEC-5 column, 500 Å, 4.6 × 300 mm (Agilent) connected to a Prominence HPLC system (Shimadzu), using running buffer A at a flow rate of 0.4 ml min$^{-1}$. The elution profiles were detected with an RF-20A fluorometer, (Shimadzu), using excitation (EX) and emission (EM) wavelengths of 433 and 475 nm for Cerulean detection, 515 nm (EX) and 528 nm (EM) for Venus detection, and 433 nm (EX) and 528 nm (EM) for FRET intensity detection, respectively[69].

**Data availability.** Coordinates and structure factors for T1r2a–T1r3LBD heterodimer with a Fab fragment (Fab16A) have been deposited in the Protein Data Bank with the accession codes 5X2M (Gln-bound), 5X2N (Ala-bound), 5X2O (Arg-bound), 5X2P (Glu-bound), and 5X2Q (Gly-bound). The sequences for the heavy and light chains of Fab16A have been deposited in the Genbank with the accession codes LC210518 (heavy chain) and LC210519 (light chain). Coordinate files of PDB IDs 1A6T (Fab fragment), 2E4U and 3SM9 (mGluR3, the extracellular domain or LBD), 1EWK, 1EWT, and 3KS9 (mGluR1LBD), 3LMK (mGluR5LBD), 3MQ4 (mGluR7LBD), 4MQE, 4MR7, and 4MS3 (GABA$_B$RLBD), 5FBK, 5K5S and 5K5T (CaSR, LBD or the extracellular domain) were used in this study. All the other data that support the findings of this study are available from the corresponding author on reasonable request.

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

## Acknowledgements

We thank Dr Yuji Ashikawa for his contribution to the early stage of this study; Drs Kazuya Hasegawa and Hideo Okumura at BL41XU, SPring-8, and Dr Naohiro Matsugaki at BL-5A, Photon Factory, for X-ray diffraction data collection support; Dr Eric Gouaux for providing the expression vector for Endo H; Dr Takao Arimori for preliminary evaluation of monoclonal antibodies; Ki-chi Hirazawa for preliminary evaluation of mutant proteins; and Fumie Iwabuki, Noriko Matsuura, Maiko Tanaka, Naoko Ono, Takashi Yamada, Emi Uchida, and Junko Nakamura for technical assistance. We also thank Drs Chikashi Toyoshima and Jian-Ren Shen for critical reading of the manuscript. The synchrotron radiation experiments were performed at the BL41XU of SPring-8, with approvals of the Japan Synchrotron Radiation Research Institute (JASRI) (Proposal No. 2013A1162, 2013B1113. 2014A1085, 2014A1859, 2014B1147, 2014B2021, 2015A1074, 2015B2074, and 2016A2534). Part of this work was supported by the Platform for Drug Discovery, Informatics, and Structural Life Science (Proposal Nos. 1034 and 1264) and Nanotechnology Platform Program (Molecule and Material Synthesis), funded by the Ministry of Education, Culture, Sports, Science and Technology, Japan. This work was financially supported by the Funding Program for Next Generation World-Leading Researchers (NEXT Program) from the Japan Society for the Promotion of Science (JSPS), the Council for Science and Technology Policy (CSTP; to A.Y.), the Targeted Proteins Research Program (TPRP; to A.Y. and Y.Ku.), Urakami Foundation for Food and Food Culture Promotion (to A.Y.), and Grant-in-Aid (Grant Number 25891017 to N.Y.) from the Ministry of Education, Culture, Sports, Science and Technology (MEXT), Japan.

## Author contributions

A.Y., N.N., N.Y., and E.N. conceived the study. N.N., N.Y., N.A. and E.N. performed protein purification. N.N., N.Y., N.A. and A.Y. performed crystallization and X-ray data collection. A.Y., N.Y., N.A., and S.A. performed the structure analysis. A.Y. and E.N. performed antibody preparation, and N.Y., Y.Ka., M.K.K., J.T., and N.N. performed antibody characterization. Y.Ku. performed the receptor assays. Y.N. performed the FRET assays. N.Y., Y.Ku., Y.N., and M.H. performed the mutation study. All authors performed data analysis. A.Y., N.N., N.Y., E.N., Y.Ku, and J.T. wrote the paper, together with input from all of the other authors.

## Additional information

**Competing interests:** The authors declare no competing financial interests.

