## [Peer Review File · Nature Communications]

Reviewer #1 (Remarks to the Author)

T1R receptors are responsible for detecting amino acid and carbohydrate taste stimuli (as well as some other natural and artificial taste stimuli that can elicit umami or sweet taste in humans), although the selectivity of the receptors can vary across vertebrate species. Here, the authors present x-ray crystallographic analyses of Medaka T1r receptor extracellular ligand binding domains in complex with various amino acid ligands. The experimental design and primary analysis appears proper, and the determination of the crystal structures of any T1R domains is an important advance. The fact that the authors were able to examine a number of amino acid ligands is important. However, there are some significant concerns regarding interpretations of the findings here that should be addressed.

1) The authors clearly imply that the results here generalize to the entire T1R family, but provide little evidence to support this conclusion. Specifically, the T1R2/T1R3 heterodimers in most mammals responsible for sweet taste primarily function in taste as mono- and disaccharide receptors (although glycine and some D-amino acids can activate them and are perceived as sweet tasting). The Medaka T1r2a receptor binds L-amino acids (and glycine), more reminiscent of the mammalian T1R1/T1R3 receptor. Therefore, it is premature to say that the results here can be generalized to orthologues, especially the mammalian sweet taste receptor. The authors should tone down their abstract and other text to acknowledge this limitation. Should the authors wish to test this possibility, more extensive molecular modelling of the rodent or human receptors along with ligand docking analyses would be helpful.

2) The authors fail to acknowledge much of the earlier work from a variety of labs that have used various approaches to understand ligand-receptor and heterodimeric interactions in the T1Rs, including work from the Margolskee, Meyerhof and Munger groups. For example, the Margolskee group has used molecular modeling to make predictions about heterodimerization in the T1Rs (2001), has mapped the aspartame binding site in the T1R2 ligand binding domain (2015), and mapped binding sites for lactisole and cyclamate in T1R3 transmembrane domains (2005; these latter studies were similarly done by the Meyerhof group). Even more relevant to this manuscript, the Munger group used spectroscopic techniques to measure saccharide binding to T1r2 and T1r3 (2005, 2006) where they found that different sugars showed different binding affinities to T1r2 but similar affinities to T1r3. Not only was none of this work cited, but the failure to do so implies a more significant advance in our understanding of the roles of two receptor subunits in ligand interactions than this paper actually is (for example, the fact that T1R3 is unlikely to confer ligand specificity is in no way unexpected, as stated in the abstract). Readers would be well served by having this work placed in the proper context of studies that have come before, especially if the authors wish to make conclusions about receptor family members outside of this one species.

Reviewer #2 (Remarks to the Author)

This paper reports a series of five crystal structures for the ligand binding domain (LBD) of a taste receptor GPCR, genetically isolated from the TM region and cys rich domain and expressed as a heterodimer. The structures are solved at sufficient resolution (2.2 - 2.6 Å) to reveal ligand, side chain and solvent density in the binding pocket. This is an important advance that for the first time gives molecular insight into how taste receptors are able to sense a broad range of ligands. In addition, the structures reveal features which underlie assembly of the T1r2a/T1r3 LBD heterodimer assembly. The paper is technically sound and will be of interest to a wide range of biologists, in part due to the novelty and biological significance of the subject. I recommend publication with high enthusiasm after appropriate revision.

The authors assume too much knowledge from readers who are not experts in the taste receptor field. Some specific examples together with additional issues which should be addressed are set out below.

Although for the most part the paper is well written, grammar could be substantially improved, and in places the text is awkward and hard to follow. In addition there are contradictory statements in the text which make no sense that add to the difficulty. Also, some of the functional data has puzzling discrepancies which the authors do not fully explain.

1) Page 2 Summary and page 3 paragraphs 1-2: To increase accessibility for the general reader it would be useful explain if T1r receptors are always obligate heterodimers, what subunit combinations are known to exist, and if ligand binding properties are conserved across species. Currently the introduction is confusing for the non expert because while human T1r2/T1r3 senses sweet, and mouse T1r1/T1r3 senses umami, fish T1r2a/T1r3 senses umami not sweet. Also the significance of subtype type a T1r2 (presumably there is another subtype) for ligand selectivity is not explained. The authors also do not do a good job of explaining current ideas about the role of the T1r3 subunit, which is important in light of their finding that it also binds amino acids.

2) Page 3 line 29: The statement that sweet and umami sensations arise from a single type of receptor is at conflict with text below explaining that these sensations are mediated by distinct molecular species. Instead of type do the authors mean a family of receptors?

3) Page 4 line 71; presumably Cys348 and Cys350 are in T1r2a? Their location is not obvious in Extended data Fig 3 but perhaps should be shown?

4) Page 4 line 62. "The structure provides a structural basis" is awkward.

5) Page 4 lines 72-76. I had difficulty in following this discussion, which does not integrate well with the explanation of disulfide bond formation earlier in the paragraph. Rewrite and reorganize the text. Having just described an intermolecular disulfide bond formed between Cys132 and Cys344, the authors then state that "No intermolecular disulfide bond is expected in T1r heterodimers".

6) Page 5 line 86 and Figure 2 legend. Explain what is being measured in panel a. Also. I do not see any evidence for a response to glutamate as written in the text on line 86, which contradicts the statement on line 89.

7) Page 8 lines 161-162: the statement that the T1r3 LBD adopts a conformation similar to the closed state is at conflict with the statement on line 24 that the pocket is more open. Rather than these descriptive statements the authors should calculate the difference in domain closure for T1r3 versus T1r2a using dyndom or a similar program; it would be useful also to report measurements for closure of T1r2a compared with mGluR and GABA receptor LBD structures.

8) Figure 2 and text on page 5 lines 84-95: what is the functional significance of the large difference in amplitude of the response to different ligands. Why does glutamate produce a FRET signal in panel b, but no change in response shown in panel a? Also, explain why the relative amplitude of the response to Gly is strikingly different in panels a and b. Finally the text on page 5 lines 88-90 is misleading since glycine (but not glutamate) does produce a response in Fig 2 panel a.

9) Figure 3 legend: there is no underlining in panel b at conflict with the text.

10) Figure 3 panel b right: the sculpting of the cavity is not helpful; perhaps try rendering the intact surface with its mapped potential as a transparent layer in photoshop, with the bound ligand shown below this layer.

11) Figure 3 panel d: move the legend to top left in this panel; currently its confusing whether the legend symbols represents data points instead.

- 12) Extended data Fig 4 legend. It would be appropriate to state that these Figures were generated with LIGPLOT.
- 13) Extended data Fig 5 legend. Explain what is being measured in panels a-d.
- 14) Extended data Fig 7 legend. Explain what is being measured in panel g.
- 15) Methods: although the Methods section is extremely detailed, and may need trimming, a major omission is any description of the purification of the T1r2a/T1r3 LBD heterodimer assembly used for crystallization in complex with a Fab fragment.
- 16) Page 19 lines 421-423. More accurately, the sugar chains were trimmed to single NAG molecules by digestion with Endo H. Although the glycosylation sites are not described (perhaps they should be mentioned in the text and EDFig1 amended to show their location) the NAG residues are clearly visible in Figure 1b.

Reviewer #3 (Remarks to the Author)

The authors analyzed the structural basis of the ligand recognition mechanism in taste receptors. They selected animal species to obtain large amount of recombinant proteins of taste receptors. In addition, they successfully prepared the specific antibody suitable for the preparation of the crystals. They got a series of structures of ligand-bound extracellular domains in taste receptors, but lack of the structure without ligand. They showed the interaction mechanism between receptor and ligand based on the structural and pharmacological results. This is a tough work and provides important information about ligand recognition in class C GPCRs. However, there are major concerns about the inconsistency of the results in the paper.

1. In p. 9, the authors discuss the relevance between the structural characteristics of ligand binding and the pharmacological results of ligand affinity (EC50). This is very important to explain the ligand recognition mechanism. Thus, the authors have to explain how to calculate the deltaG values from the structures and the EC50 values.
2. In Fig. 2c and EDFig. 2b, the overall structures of LBD are almost the same, irrespective of binding amino acids. Thus, the distances between C-terminals of LBDs are quite similar in these structures. However, in Fig. 2b, the maximum values of FRET index are different among amino acids. Especially, the max value of Gly binding is much larger than that of Gln binding. The authors have to explain this inconsistency.
3. In class C GPCRs, the dimeric ECDs approach to each other, which induces the G protein activation in the transmembrane domains. However, in Fig. 2a,b, the max values of the elevation of Ca²⁺ elevation are not correlated with the max values of FRET index. The authors have to explain the relationship between the Ca²⁺ response and the structural change of ECD under saturated ligand concentration.
4. In EDFig. 6b, the authors show the dimerization of the fusions between LBD and fluorescence protein. The monomers are ~100kDa, but the dimer is much larger than 250kDa. In EDig. 3, the size of the LBD dimer is reasonable based on the size of the monomers. I am concerned about some effects of the fluorescence protein fusion on the dimerization.

Minor points

1. The manuscript contains "T1r3", "T1R3" and "T13" for the same meaning. Standardization of words is helpful.

2. P. 3, L 13; "has been has been" should be revised as "has been".
3. Legend of Fig. 3d; closed and open symbols are not used in the figure.

Responses to reviewers' comments

Reviewer #1 (Remarks to the Author):

T1R receptors are responsible for detecting amino acid and carbohydrate taste stimuli (as well as some other natural and artificial taste stimuli that can elicit umami or sweet taste in humans), although the selectivity of the receptors can vary across vertebrate species. Here, the authors present x-ray crystallographic analyses of Medaka T1r receptor extracellular ligand binding domains in complex with various amino acid ligands. The experimental design and primary analysis appears proper, and the determination of the crystal structures of any T1R domains is an important advance. The fact that the authors were able to examine a number of amino acid ligands is important.

Reply: We thank the reviewer for his/her appreciation of our work.

However, there are some significant concerns regarding interpretations of the findings here that should be addressed.

1) The authors clearly imply that the results here generalize to the entire T1R family, but provide little evidence to support this conclusion. Specifically, the T1R2/T1R3 heterodimers in most mammals responsible for sweet taste primarily function in taste as mono- and disaccharide receptors (although glycine and some D-amino acids can activate them and are perceived as sweet tasting). The Medaka T1r2a receptor binds L-amino acids (and glycine), more reminiscent of the mammalian T1R1/T1R3 receptor. Therefore, it is premature to say that the results here can be generalized to orthologues, especially the mammalian sweet taste receptor. The authors should tone down their abstract and other text to acknowledge this limitation. Should the authors wish to test this possibility, more extensive molecular modelling of the rodent or human receptors along with ligand docking analyses would be helpful.

Reply: We are of the opinion that fish T1r2s are rather good models for both T1rs and T1r2s of mammals since in terms of sequence identity they are equally similar to both (Ref # 26, Ishimaru *et al.* 2005). We have included sentences to this effect in the last paragraph in the

Introduction (p.4). However we also acknowledge that generalizing to all T1rs may be premature and have toned down the Abstract and Discussion in this regard as suggested.

2) The authors fail to acknowledge much of the earlier work from a variety of labs that have used various approaches to understand ligand-receptor and heterodimeric interactions in the T1Rs, including work from the Margolskee, Meyerhof and Munger groups. For example, the Margolskee group has used molecular modeling to make predictions about heterodimerization in the T1Rs (2001), has mapped the aspartame binding site in the T1R2 ligand binding domain (2015), and mapped binding sites for lactisole and cyclamate in T1R3 transmembrane domains (2005; these latter studies were similarly done by the Meyerhof group). Even more relevant to this manuscript, the Munger group used spectroscopic techniques to measure saccharide binding to T1r2 and T1r3 (2005, 2006) where they found that different sugars showed different binding affinities to T1r2 but similar affinities to T1r3. Not only was none of this work cited, but the failure to do so implies a more significant advance in our understanding of the roles of two receptor subunits in ligand interactions than this paper actually is (for example, the fact that T1R3 is unlikely to confer ligand specificity is in no way unexpected, as stated in the abstract). Readers would be well served by having this work placed in the proper context of studies that have come before, especially if the authors wish to make conclusions about receptor family members outside of this one species.

Reply: We thank the reviewer for his/her important comment. We were limited by the number of allowed references, and have addressed the concerns as follows:

Heterodimerization – We acknowledge that Margolskee considered the possibility of heterodimerization in 2002 (*JBC* **277**, 1), based on a homology model of T1r3 homodimer reported in 2001 (Max *et al. Nature Genetics*, **28**, 58). However, we also acknowledge that the possibility of the heterodimerization itself was already addressed in an earlier study by Nelson *et al.* in 2001 (*Cell*, **106**, 381) with experimental evidence, and further verified experimentally in 2002 (*Nature*, **416**, 199), the ones we had used in our manuscript. Since we now have a maximum number of the references because of the additions including the following ones, we have kept the references regarding the heterodimerization to these two.

Modeling and prediction of ligand binding sites – We have added several modeling/docking studies, including Maillet *et al.* (*Chem Senses*, **40**, 577, 2015) and Jiang *et al.* (*JBC* **280**, 34296, 2005).

Ligand binding to T1r3 – We now refer to the Munger group (Nie *et al. Curr. Biol.* **15**, 1948, 2005) and the Briand group (Maîtrepierre, *et al. Prot, Exp. Purif.* **83**, 75, 2012), that reported recombinant preparations of T1r3LBD and analyzed binding of sweet substances. Regarding the ligand binding assay by T1r2LBD, we found that only one group reported using the fusion protein with maltose binding protein (Nie *et al.* 2005), and the observed fluorescent change was very small compared to generally observed values using this method. Therefore we rather refrain from discussing the relative affinities of T1r2 and T1r3 in detail based on the values reported in the reference.

Reviewer #2 (Remarks to the Author):

This paper reports a series of five crystal structures for the ligand binding domain (LBD) of a taste receptor GPCR, genetically isolated from the TM region and cys rich domain and expressed as a heterodimer. The structures are solved at sufficient resolution (2.2 - 2.6 Å) to reveal ligand, side chain and solvent density in the binding pocket. This is an important advance that for the first time gives molecular insight into how taste receptors are able sense a broad range of ligands. In addition the structures reveal features which underlie assembly of the T1r2a/T1r3 LBD heterodimer assembly. The paper is technically sound and will be of interest to a wide range of biologists, in part due to the novelty and biological significance of the subject. I recommend publication with high enthusiasm after appropriate revision.

Reply: We thank the reviewer for his/her appreciation of our work.

The authors assume too much knowledge from readers who are not experts in the taste receptor field. Some specific examples together with additional issues which should be addressed are set out below.

Although for the most part the paper is well written, grammar could be substantially improved, and in places the text is awkward and hard to follow. In addition there are contradictory statements in the text which make no sense that add to the difficulty. Also, some of the functional data has puzzling discrepancies which the authors do not fully explain.

Reply: The Introduction has been revised to provide more background information. The manuscript has been edited by a scientific English editing service. The other specific points we revised are described in the following point-by-point response section.

1) Page 2 Summary and page 3 paragraphs 1-2: To increase accessibility for the general reader it would be useful explain if T1r receptors are always obligate heterodimers, what subunit combinations are known to exist, and if ligand binding properties are conserved across species. Currently the introduction is confusing for the non expert because while human T1r2/T1r3 senses sweet, and mouse T1r1/T1r3 senses umami, fish T1r2a/T1r3 senses umami not sweet. Also the significance of

subtype type a T1r2 (presumably there is another subtype) for ligand selectivity is not explained. The authors also do not do a good job of explaining current ideas about the role of the T1r3 subunit, which is important in light of their finding that it also binds amino acids.

Reply: We revised the Introduction accordingly and added information regarding heterodimerization, ligand specificity of the T1r receptors including fish T1r2/T1r3 receptors, and the ligands reportedly targeting T1r3 subunit, and briefly addressed possible roles of T1r3 in the first paragraph in the Discussion.

2) Page 3 line 29: The statement that sweet and umami sensations arise from a single type of receptor is at conflict with text below explaining that these sensations are mediated by distinct molecular species. Instead of type do the authors mean a family of receptors?

Reply: The Introduction has been revised to clarify what we meant by broad ligand specificity of T1rs.

3) Page 4 line 71; presumably Cys348 and Cys350 are in T1r2a? Their location is not obvious in Extended data Fig 3 but perhaps should be shown?

Reply: Cys348 and Cys350 are in T1r2a (Supplementary Figure 1), but we have decided to remove this potentially confusing sentence.

4) Page 4 line 62. "The structure provides a structural basis" is awkward.

Reply: We have modified the sentence.

5) Page 4 lines 72-76. I had difficulty in following this discussion, which does not integrate well with the explanation of disulfide bond formation earlier in the paragraph. Rewrite and reorganize the text. Having just described an intermolecular disulfide bond formed between Cys132 and Cys344, the authors then state that "No intermolecular disulfide bond is expected in T1r heterodimers".

Reply: We have rewritten the text.

6) Page 5 line 86 and Figure 2 legend. Explain what is being measured in panel a. Also. I do not see any evidence for a response to glutamate as written in the text on line 86, which contradicts the statement on line 89.

Reply: Legend to the current Figure 2b has been amended, and the relevant text has been revised (the first paragraph in the “Ligand recognition by T1r2a subunit”, p.6).

7) Page 8 lines 161-162: the statement that the T1r3 LBD adopts a conformation similar to the closed state is at conflict with the statement on line 24 that the pocket is more open. Rather than these descriptive statements the authors should calculate the difference in domain closure for T1r3 versus T1r2a using dyndom or a similar program; it would be useful also to report measurements for closure of T1r2a compared with mGluR and GABA receptor LBD structures.

Reply: According to the reviewer’s suggestion, we have performed the domain motion analyses by DynDom, and the results are described in the second paragraph of the “Ligand binding at T1r3 subunit” subsection (p. 10) for T1r3LBD and in the second paragraph of the “Ligand recognition by T1r2a subunit” subsection (p. 7) for T1r2aLBD. A summary is provided in Supplementary Table 3.

8) Figure 2 and text on page 5 lines 84-95: what is the functional significance of the large difference in amplitude of the response to different ligands. Why does glutamate produce a FRET signal in panel b, but no change in response shown in panel a? Also, explain why the relative amplitude of the response to Gly is strikingly different in panels a and b. Finally the text on page 5 lines 88-90 is misleading since glycine (but not glutamate) does produce a response in Fig 2 panel a.

Reply: We have added a paragraph at the end of the Discussion explaining current thinking on why there are different maximum FRET to different ligands. It is based on the idea that there are multiple conformational states, not just “active” and “inactive”, that are variously populated at ligand saturation according to ligand identity.

In addition, it should be noted that the dose-response assay for glycine and glutamate by Ca^{2+} -imaging were not investigated up to their expected saturated concentrations, in order to avoid non-specific reactions aroused by addition of high

concentration chemicals (We set the ligand concentration for the assay up to 10mM; please see a supplementary figure below). Therefore, since we did not conclude what kinds of effects are actually exerted on the receptor by the binding of glycine and glutamate to LBD, we have defined that the broad chemical specificity dealt with in this study is that for recognition by LBD. We have added the information and rewrote the text in the first paragraph in the “Ligand recognition by T1r2a subunit” subsection in the Result (p.6).

A preliminary experiment to determine the ligand concentration range for the dose-response assay of the full-length T1r2a/T1r3 receptor in HEK293 cells by Ca²⁺-imaging. Because ligand application beyond 10 mM resulted in reduced signals irrespective of amino acids under our experimental condition, possibly due to chemical toxicity, we set the ligand concentration range up to 10 mM for the experiments reported in the paper.

9) Figure 3 legend: there is no underlining in panel b at conflict with the text.

Reply: We have corrected the text.

10) Figure 3 panel b right: the sculpting of the cavity is not helpful; perhaps try rendering the intact surface with its mapped potential as a transparent layer in photoshop, with the bound ligand shown below this layer.

Reply: The interior surface representation of the cavity in Figure 3 panel b is prepared in the same way as in Figure 2 panel j and k to facilitate comparison using the program HOLLOW. HOLLOW provides the Connolly surface interior of the protein (Ho & Gruswitz, *BMC Struct Biol* **8**, 49, 2008), which avoids confusion with the exterior surface. We would like to leave the rendition as it is.

11) Figure 3 panel d: move the legend to top left in this panel; currently its confusing whether the legend symbols represents data points instead.

Reply: We have revised the panel.

12) Extended data Fig 4 legend. It would be appropriate to state that these Figures were generated with LIGPLOT.

Reply: We have added the information, in the current Supplementary Figure 5 legend.

13) Extended data Fig 5 legend. Explain what is being measured in panels a-d.

Reply: We have added the information, in the current Supplementary Figure 6 legend.

14) Extended data Fig 7 legend. Explain what is being measured in panel g.

Reply: We have added the information, in the current Supplementary Figure 8 legend.

15) Methods: although the Methods section is extremely detailed, and may need trimming, a major omission is any description of the purification of the T1r2a/T1r3 LBD heterodimer assembly used for crystallization in complex with a Fab fragment.

Reply: We have trimmed the Methods section. Now the Methods section has 2,132 words and is within the limitation (max. 3000 words), even after the addition of the several information according to the journal's guideline,.

16) Page 19 lines 421-423. More accurately, the sugar chains were trimmed to single NAG molecules by digestion with Endo H. Although the glycosylation sites are not described (perhaps they should be mentioned in the text and EDFig1 amended to show their location) the NAG residues are clearly visible in Figure 1b.

Reply: The text has been modified appropriately (p.6). We mention the glycosylation observed in the crystal structure in the legend of Figure 1b, and added information about the glycosylation sites in Supplementary Fig. 1.

Reviewer #3 (Remarks to the Author):

The authors analyzed the structural basis of the ligand recognition mechanism in taste receptors. They selected animal species to obtain large amount of recombinant proteins of taste receptors. In addition, they successfully prepared the specific antibody suitable for the preparation of the crystals. They got a series of structures of ligand-bound extracellular domains in taste receptors, but lack of the structure without ligand. They showed the interaction mechanism between receptor and ligand based on the structural and pharmacological results.

This is a tough work and provides important information about ligand recognition in class C GPCRs.

Reply: We thank the reviewer for his/her appreciation of our work.

However, there are major concerns about the inconsistency of the results in the paper.

1. In p. 9, the authors discuss the relevance between the structural characteristics of ligand binding and the pharmacological results of ligand affinity (EC50). This is very important to explain the ligand recognition mechanism. Thus, the authors have to explain how to calculate the ΔG values from the structures and the EC50 values.

Reply: We already had the explanations how to calculate the ΔG values for Figure 3d in the Methods section, in the last sentence of the first paragraph in the “Measurement of the response of T1r2 and T1r3 to ligands” subsection (p. 13) and in the second last sentence of the last paragraph in the “Crystallography” subsection (p. 18). However, for further clarification, we added some more details.

2. In Fig. 2c and ED Fig. 2b, the overall structures of LBD are almost the same, irrespective of binding amino acids. Thus, the distances between C-terminals of LBDs are quite similar in these structures. However, in Fig. 2b, the maximum values of FRET index are different among amino acids. Especially, the max value of Gly binding is much larger than that of Gln binding. The authors have to explain this inconsistency.

Reply: We have added a discussion about the inconsistency between observed T1r structures and FRET changes as a last paragraph in the Discussion section. It is based on the idea that there are multiple conformational states, not just “active” and “inactive”, that are variously populated at ligand saturation according to ligand identity. In this model, one presumes that crystallization only traps one of the conformational states.

3. In class C GPCRs, the dimeric ECDs approach to each other, which induces the G protein activation in the transmembrane domains. However, in Fig. 2a,b, the max values of the elevation of Ca²⁺ elevation are not correlated with the max values of FRET index. The authors have to explain the relationship between the Ca²⁺ response and the structural change of ECD under saturated ligand concentration.

Reply: The discussion about the relationship between structural change of ECD and receptor responses has been also added in the last paragraph in the Discussion section.

As we wrote above, we consider that the different max values for FRET may reflect different occupancies of multiple conformational states of T1rLBD. In addition, it should be noted that the dose-response assay for glycine and glutamate by Ca²⁺-imaging were not investigated up to their expected saturated concentrations, in order to avoid non-specific reactions aroused by addition of high concentration chemicals (We set the ligand concentration for the assay up to 10mM; please see a supplementary figure below). Therefore, since we did not conclude what kinds of effects are actually exerted on the receptor by the binding of glycine and glutamate to LBD, we have defined that the broad chemical specificity

A preliminary experiment to determine the ligand concentration range for the dose-response assay of the full-length T1r2a/T1r3 receptor in HEK293 cells by Ca²⁺-imaging. Because ligand application beyond 10 mM resulted in reduced signals irrespective of amino acids under our experimental condition, possibly due to chemical toxicity, we set the ligand concentration range up to 10 mM for the experiments reported in the paper.

dealt with in this study is that for recognition by LBD. We have added the information and rewrote the text in the first paragraph in the “Ligand recognition by T1r2a subunit” subsection in the Result (p.6).

4. In EDFig. 6b, the authors show the dimerization of the fusions between LBD and fluorescence protein. The monomers are ~100kDa, but the dimer is much larger than 250kDa. In EDig. 3, the size of the LBD dimer is reasonable based on the size of the monomers. I am concerned about some effects of the fluorescence protein fusion on the dimerization.

Reply: As explained in the legend for the current Supplementary Fig. 7c, the estimated molecular weight values for the top of the peak of the samples were ~ 240 kDa, while the calculated value for T1r2aLBD-Celurean/T1r3LBD-Venus heterodimer is ~180 kDa. We consider that the 33% discrepancy is acceptable, taking into account that the proteins consist of multiple domains (T1rLBD and the fluorescent protein) and have multiple sugar chains, which would tend to give a larger hydrodynamic radius compared to those for globular proteins.

Minor points

1. The manuscript contains “T1r3”, “T1R3” and “T13” for the same meaning. Standardization of words is helpful.
2. P. 3, L 13; “has been has been“ should be revised as “has been”.
3. Legend of Fig. 3d; closed and open symbols are not used in the figure.

Reply: We corrected all these points.

Reviewer #1 (Remarks to the Author)

The authors have adequately addressed my concerns.

Reviewer #2 (Remarks to the Author)

The revised manuscript is substantially improved, especially with respect to making the study accessible to a non specialist audience, and also in clarifying the interpretation of the structural and functional studies. This is an important piece of work which advances our understanding of the molecular mechanisms underlying taste sensation, by providing the first experimentally determined structures of taste receptor LBDs.

Although substantially improved a few grammatical errors remain, which hopefully can be corrected by the copy editors.

There is one statement in the methods that does need clarification and revision: the meaning of the phrase 'technical replicate' is unclear. Does this mean that the experiment was repeated independently, or that in one experiment counts from multiple sample wells were averaged, and so on?

Reviewer #3 (Remarks to the Author)

The authors appropriately dealt with most points I raised in my previous review. I have not further points at this stage. I recommend publication for the journal.

Responses to Referee

Reviewer #2 (Remarks to the Author):

The revised manuscript is substantially improved, especially with respect to making the study accessible to a non specialist audience, and also in clarifying the interpretation of the structural and functional studies. This is an important piece of work which advances our understanding of the molecular mechanisms underlying taste sensation, by providing the first experimentally determined structures of taste receptor LBDs.

Reply: We thank the reviewer for his/her appreciation of our work.

Although substantially improved a few grammatical errors remain, which hopefully can be corrected by the copy editors.

Reply: We subjected the final version of the manuscript to grammar check once again, and made corrections accordingly.

There is one statement in the methods that does need clarification and revision: the meaning of the phrase 'technical replicate' is unclear. Does this mean that the experiment was repeated independently, or that in one experiment counts from multiple sample wells were averaged, and so on?

Reply: We clarified this point in the last paragraph of the subsection “Measurement of the response of T1r2 and T1r3 to ligands” in the Methods section (p. 14).